# In-plane charged antiphase boundary and 180° domain wall in a ferroelectric film

Xiangbin Cai [1,2,10] ✉, Chao Chen[1,10], Lin Xie [3], Changan Wang[4,5], Zixin Gui[1], Yuan Gao[6], Ulrich Kentsch [4], Guofu Zhou[1], Xingsen Gao [1], Yu Chen[7], Shengqiang Zhou [4], Weibo Gao [2], Jun-Ming Liu [1,8], Ye Zhu[9] & Deyang Chen [1] ✉

The deterministic creation and modification of domain walls in ferroelectric films have attracted broad interest due to their unprecedented potential as the active element in non-volatile memory, logic computation and energy-harvesting technologies. However, the correlation between charged and antiphase states, and their hybridization into a single domain wall still remain elusive. Here we demonstrate the facile fabrication of antiphase boundaries in $BiFeO_3$ thin films using a He-ion implantation process. Cross-sectional electron microscopy, spectroscopy and piezoresponse force measurement reveal the creation of a continuous in-plane charged antiphase boundaries around the implanted depth and a variety of atomic bonding configurations at the anti-phase interface, showing the atomically sharp 180° polarization reversal across the boundary. Therefore, this work not only inspires a domain-wall fabrication strategy using He-ion implantation, which is compatible with the wafer-scale patterning, but also provides atomic-scale structural insights for its future utilization in domain-wall nanoelectronics.

In comparison to the three-dimensional domains in ferroelectric thin films, the two-dimensional domain walls or boundaries, which separate domains of distinct crystallographic and electronic features, typically have the reduced dimensionality of only a few atomic layers. These domain walls possess broken local symmetry and perturbed chemical environments, rendering them with promising aspects for nanoelectronics and nano-optoelectronics[1–8]. A wealth of physico-chemical properties have been discovered in ferroelectric domain walls, including domain-wall conductivity[9–15], unconventional magnetism[16], optical response[17,18], giant electromechanical[19] and thermotropic behaviors[20].

These domain walls can be categorized as either electrically neutral or charged by the net bound charge accumulated from the normal polarization of neighboring domains[21]. Specifically, the spontaneous polarization in ferroelectric materials here is defined as the dipole vector drawing from the negative to positive centers of each unit cell (u.c.), and thus in general a domain wall in the head-to-head (or tail-to-tail) state tends to be positively (or negatively) charged, while the defect types accumulating around may affect the charge states. In ferroelectric thin films with out-of-plane polarization, it is normal to observe the neutral domain wall elongating along the

[1]Guangdong Provincial Key Laboratory of Optical Information Materials and Technology, and Institute for Advanced Materials, South China Academy of Advanced Optoelectronics, South China Normal University, Guangzhou 510006, China. [2]Division of Physics and Applied Physics, School of Physical and Mathematical Sciences, Nanyang Technological University, Singapore 637371, Singapore. [3]Department of Physics, Southern University of Science and Technology, Shenzhen 518055, China. [4]Helmholtz-Zentrum Dresden-Rossendorf, Institute of Ion Beam Physics and Materials Research, Dresden 01328, Germany. [5]School of Electronics & Communication, Guangdong Mechanical and Electrical Polytechnic, Guangzhou 510515, China. [6]State Key Laboratory of Nuclear Physics and Technology, School of Physics, Peking University, Beijing 100871, China. [7]Institute of High Energy Physics, Chinese Academy of Sciences, Beijing 100049, China. [8]Laboratory of Solid State Microstructures and Innovation Center of Advanced Microstructures, Nanjing University, Nanjing 210093, China. [9]Department of Applied Physics, Research Institute for Smart Energy, The Hong Kong Polytechnic University, Hong Kong, China. [10]These authors contributed equally: Xiangbin Cai, Chao Chen. ✉e-mail: xcaiak@connect.ust.hk; deyangchen@m.scnu.edu.cn

polarization direction, while a charged domain wall extending perpendicular to the film polarization direction is barely found in as-grown films (although examples exist in capacitor geometries[22]), because of the energetically unfavored larger area-to-body ratio and the extra electrostatic energy built up by charging. However, such in-plane charged domain walls are appealing by their non-trivial charged properties and functionalities of more viable controllability, which are unavailable by out-of-plane domain walls, such as the ferroelectric tunnel junction displaying discrete quantum-well energy levels[23]. Recently, Liu et al. also fabricated the smallest memristor of multiple resistance states facilitated by the layer-by-layer migration of single in-plane charged domain wall in a 4 u.c. BiFeO₃ (BFO) film[24]. Such fascinating quasi-two-dimensional nature of in-plane charged domain walls, in regardless of the film thickness, may enable the design of the electronic circuitry at a sub-nano scale.

On the other hand, the domain walls in ferroelectric thin films can be distinguished in respect of their crystallographic geometries, such as antiphase[25,26] and twin[27] boundaries. In contrast to the twin boundaries involving a reflection of neighboring lattices, the antiphase boundaries feature a planar defect by translating half-unit-cell of atomic registry in reference to the adjacent domain. In addition to the competing effects (either serving as nucleation sites[28] or promoting the retention failure[29]) of antiphase boundaries during the BFO domain switching, they also play critical roles in the anomalous magnetoresistance of Fe₃O₄[30] and the antiferromagnetic coupling of NiFe₂O₄[31]. Since BFO holds robust magnetoelectric coupling above the room temperature, the hybridization of both in-plane charged (i.e., electrically switchable) and antiphase characteristics into a single-domain wall seems a charming idea and may pave way to the possibility of controlling magnetism with electric field at room temperature[32,33]. However, the realization and integration of such in-plane charged antiphase boundary (IP-CAPB) in ferroelectric films has been rather challenging and, to our best knowledge, not reported yet.

Since the antiphase boundary physically requires the half-unit-cell shift of atomic registry across the boundary, its generation poses tough conditions on the technique to be used. As a technique of tunable energy range, being compatible with the large-scale patterning and reversible by thermal annealing[34], the He-ion implantation treatment has been widely applied to a multitude of materials to engineer their microscopic structure and apparent properties to high precision. For example, the negatively charged boron vacancy ($V_B^-$) spin defects in hexagonal boron nitride was synthesized by the He-ion implantation

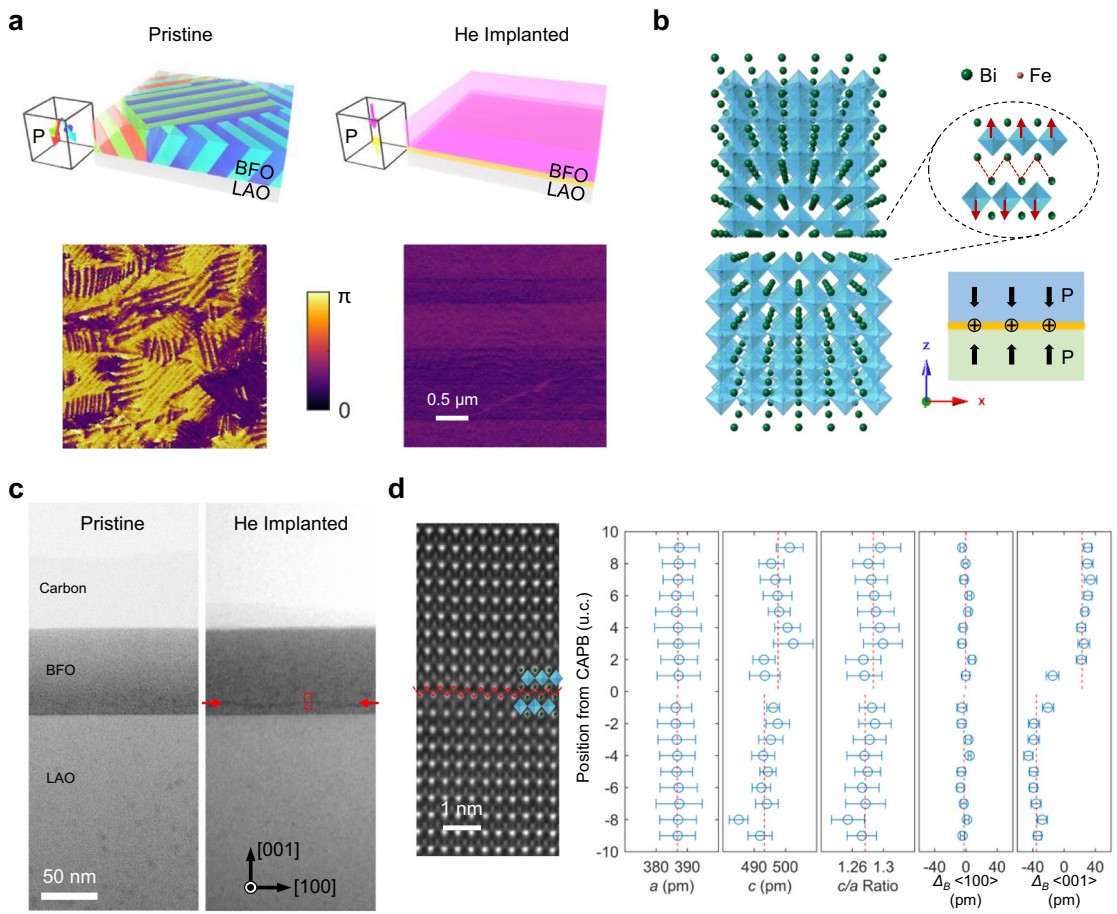

**Fig. 1 | Fabrication of in-plane charged antiphase boundaries (IP-CAPBs) in BiFeO₃ (BFO) thin films. a** Schematics illustrating the domain structures in BFO thin films on LaAlO₃ (LAO) substrates before and after the He-ion implantation. The LAO substrates are in gray while the different domain colors in BFO films before and after the He-ion implantation represent various polarization vectors as sketched in corresponding cuboids on the left. The respective in-plane piezoresponse force microscopy (IP-PFM) measurements are presented below. **b** Calculated atomic models of one IP-CAPB in the BFO matrix, demonstrating spontaneous head-to-head polarization initialized around the domain wall. The 2D-projected polarization of one BFO unit cell (u.c.) near the CAPB can be represented by the displacement vector from the Fe atomic column relative to the center of corresponding u.c. (the center of Bi rectangle). **c** Cross-sectional annular bright field (ABF) imaging of the BFO/LAO films before and after the He-ion implantation. The red dashed rectangle region is zoomed in as (**d**) by high-angle annular dark field (HAADF) imaging for more accurate determination of atomic positions, where the red zigzag line marks the antiphase interface, and the extracted lattice parameters are statistically analyzed on the right. The error bars denote the standard deviation of corresponding dataset from each u.c. row while the red dashed lines indicate the mean values of regions above and beneath the CAPB.

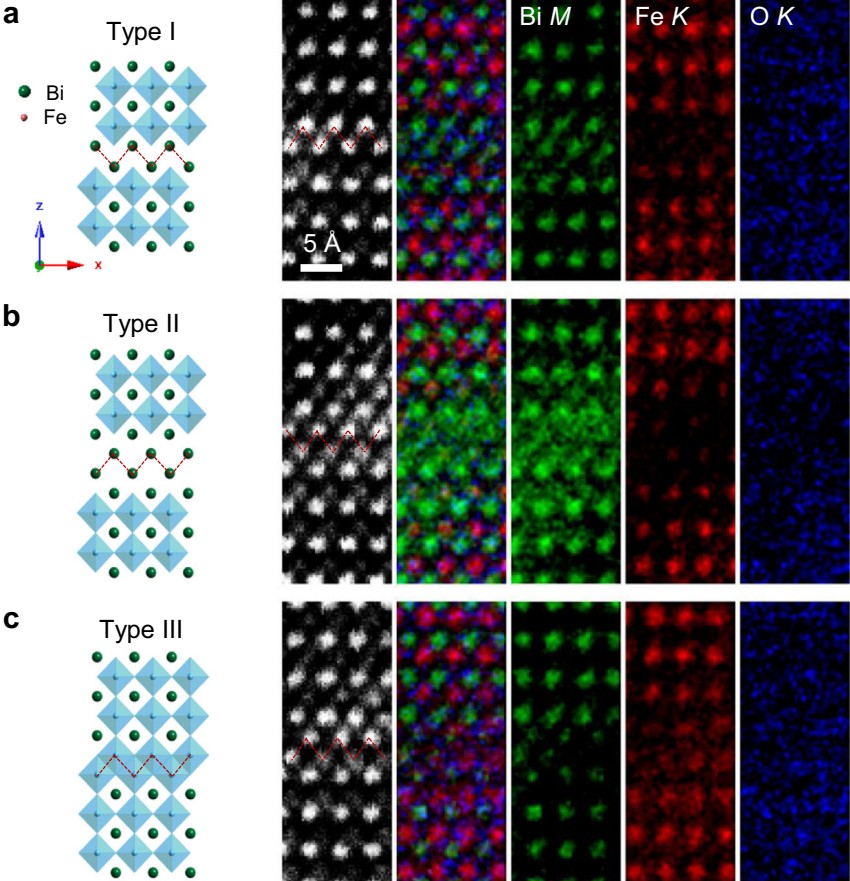

**Fig. 2 | Different types of IP-CAPBs found in He-implanted BFO thin films.**
Atomic models, simultaneous HAADF images, overlaid composition maps and the constituent sub-maps by integrating Bi-*M*, Fe-*K*, O-*K* energy-dispersive X-ray spectroscopy (EDS) peaks, respectively, viewed along the <010> zone axis, are presented from left to right for **a** type-I, **b** type-II and **c** type-III CAPBs. We notice that the type-III CAPB involves the direct face-to-face connection of FeO$_6$ octahedra and thus exhibits a high-energy state that is spatially discontinuous. The red zigzag lines mark the antiphase interfaces in respective configurations.

to facilitate the coherent control of nitrogen nuclear spins at room temperature[35]. It has also been developed for ferroelectric films to produce the single-domain super-tetragonal phase in BFO[36] and to triple the energy storage density in PbZrO$_3$ by breaking film polarization into polar nanoregions[37], though its implementation in the domain-wall generation remains largely unexplored.

In this paper, we report the facile construction of a spatially continuous IP-CAPB in the BFO thin film using a He-ion implantation technique. The ferroelectric-domain-modification effect of He implantation is demonstrated by comparing the piezoresponse force microscopy (PFM) and cross-sectional TEM results of the same BFO/LAO film before and after the He-implantation treatment. PFM measurements reveal that the in-plane multi-domain structure is refined into a uniform out-of-plane single-domain polarization near the surface of films. Cross-sectional scanning transmission electron microscopy (STEM) and atomic-resolution energy-dispersive X-ray spectroscopy (EDS) together elucidate various configurations of atomic bonding at the antiphase interface, which interweave to form an IP-CAPB around the implanted depth. Through quantitative analysis of atomic column positions, the atomically sharp 180° polarization reversal across the IP-CAPB is identified. We also discover a single-unit-cell thick antipolar Bi$_2$O$_3$ inside the bending region of CAPB, which switches the nearest single-unit-cell BFO layer to form multiple charged interfaces, which relieve the electrostatic energy and stabilize the IP-CAPB. Our work thus sheds light on the domain-wall engineering by He-ion implantation and provides the direct atomic-scale observation of such unique IP-CAPBs for their future

utilization in domain-wall nanoelectronics, spintronics and memory device applications.

## Results and discussion

We grew BFO films of ~70 nm thickness on LaAlO$_3$ (LAO) (001) substrates using the pulsed laser deposition (PLD) technique as detailed in the Method section. The pristine BFO film is dominated by the tetragonal-like (M$_C$) phase as indicated by the X-ray diffraction (XRD) results in Supplementary Fig. S1, exhibiting the strong (002) M$_C$ (tetragonal-like phase) and weak (002) M$_A$ (rhombohedral-like phase) peaks near the (002) LAO substrate peak[38–41]. The topography measurement by atomic force microscopy (AFM) in Supplementary Fig. S2a shows some dark patch contrast, which can be ascribed to the rhombohedral-like (R-like) phase naturally mixed into the M$_C$ matrix when the film thickness exceeds ~40 nm and the substrate constraint relieves[38,39]. Because R-like phases come from the distortion of parent matrix to exhibit a smaller c parameter, manifesting lower local height in the topographic AFM map[38,39]. As shown in the schematics and the in-plane piezoresponse force microscopy (IP-PFM) phase maps in Fig. 1a, the as-grown BFO film contains stripe domains of distinct in-plane polarization components, evincing a phase difference of π (180°). After the characterization, the BFO/LAO heterostructure was bombarded by 8 kV He ions with a fluence of $2.5 \times 10^{15}$ ions per cm$^2$. It can be seen that the in-plane domain structures disappear after the He-ion implantation, leaving a uniform IP-PFM contrast in Fig. 1a (some horizontal stripes of tiny phase variation come from flyback artifacts in the slow scan direction). By

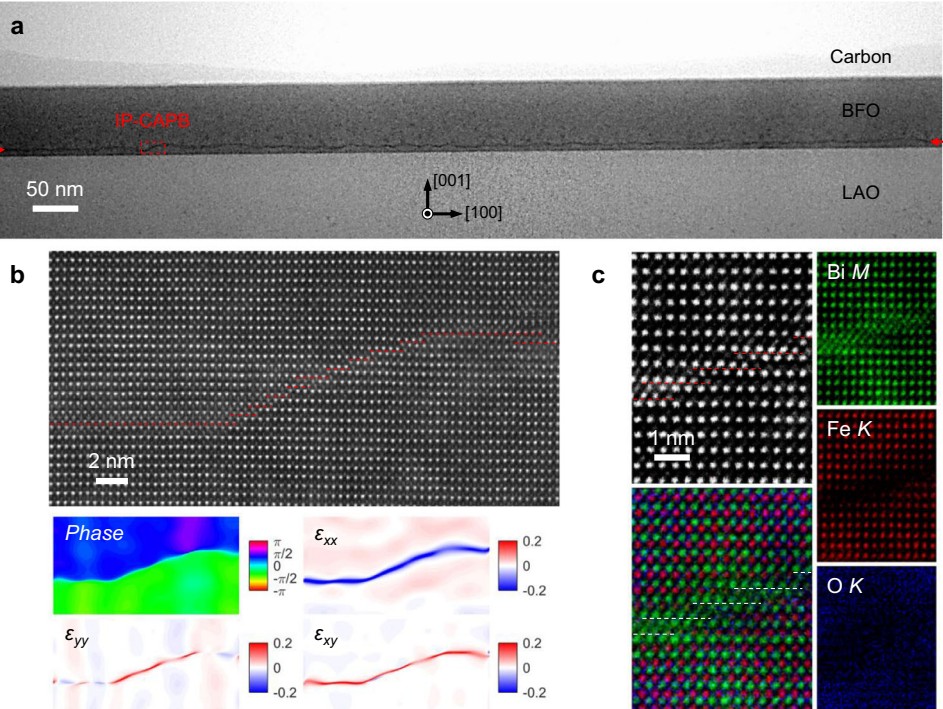

**Fig. 3 | IP-CAPB by the cooperative interweaving of antiphase configurations.**
**a** Low-magnification ABF image showing a continuous CAPB across the entire BFO thin film. One of the bending points is enlarged in (**b**) for the atomic HAADF observation. Geometric phase analysis (GPA) is applied to extract the phase and strain distribution in this bending area as shown in lower panels. **c** Atomic-resolution EDS mapping reveals that the bending behavior is enabled by the cooperative interchange between type-I and type-II CAPBs of few-unit-cell lengths. The type-II CAPB is formed naturally in the overlapped region of two closest neighboring type-I CAPBs.

XRD in Supplementary Fig. S1 and reciprocal space map (RSM) in Supplementary Fig. S5, we notice that the BFO film transforms into the true tetragonal (T) phase after the He-ion implantation. This phase transition is consistent with the AFM topography measurement in Supplementary Fig. S2b, where the interweaving bright-dark stripe domain contrast is gone, and the atomic terraces on the surface can be observed. The out-of-plane (OOP) PFM phase maps of samples before and after the He-ion implantation, as shown in Supplementary Fig. S2, do not exhibit any phase reversal or detectable domain contrast, suggesting that the He-ion implantation process did not alter the OOP polarization component near the surface of films[42]. Due to the insulating nature of LAO substrates, OOP-PFM signal is a bit noisy. However, the homogeneous contrast in both IP- and OOP-PFM results indicates the single-domain state near the surface of BFO films after implantation.

According to the motivation as discussed previously in the introduction paragraph to hybridize the antiphase and the in-plane charged characteristics into a single-domain wall, we simulated the possible atomic structure by artificially constructing one antiphase interface in the BFO matrix. After geometric and energy refinements, there is spontaneous head-to-head polarization initialized in BFO around the boundary as shown in Fig. 1b, which is represented by the displacement of Fe atomic columns relative to the corresponding u.c. centers (the centers of Bi rectangles). Owing to the head-to-head OOP polarization aside the antiphase interface, the CAPB is positively charged by bound charges, since we did not observe possible sources of screening electrons. Such atomic model is verified further by direct atomic-scale observation. Using focused ion beam (FIB) techniques, we lifted out cross-sectional membranes from the He-ion implanted BFO/LAO films for the structure characterization by scanning transmission electron microscopy (STEM), as the scanning electron microscopy (SEM) image in Supplementary Fig. S3 shows. The BFO membrane is

cut out along its <100> direction and viewed along its <010> zone axis. To avoid surface damage during the milling processes, the BFO is capped with both carbon paint and deposited Pt protection layer. In Fig. 1c, the cross-sectional annular bright field (ABF) images of the BFO/LAO films before and after the He-ion implantation show a difference in the occurrence of a single dark line at ~60 nm depth of the processed BFO film. This depth position is in consistence with our simulation of He implantation depth profiles as shown in Supplementary Fig. S6. The accumulation and migration of oxygen vacancies and generated anti-site defects to a critical level by the He implantation compose the IP-CAPB at the implantation front, where the high-enough He-ion density can arrive under 8 kV beam energy. Varied beam energies can lead to different implantation depths and profiles as shown in Supplementary Fig. S6. Thus a different IP-CAPB position can be expected if a different energy is used for the implantation. A zoomed-in view by high-angle annular dark field (HAADF) imaging in Fig. 1d demonstrates the exact atomic configurations of antiphase interface (marked by red zigzag lines) in high consistency with the calculated atomic model. Through fitting the HAADF intensity of atomic columns with 2D Gaussian functions, we are able to extract the atomic coordinates and calculate the lattice parameters unit cell by unit cell[24]. The average and standard deviation values of every u.c. row are plotted as the function of distance away from the CAPB on the right panel of Fig. 1d, by which we can see the evident change of c/a ratios across the CAPB in agreement with the split (002) sub-peaks in XRD. The <100> polarization component is nearly zero while the <001> component switches its sign across the interface, indicating an atomically sharp head-to-head 180° domain wall, instead of the gradual rotation of polarization across the reported in-plane charged domain wall[24]. The cross-sectional microscopic analysis of the BFO/LAO interface can be seen in Supplementary Fig. S4, demonstrating the single-domain polarization of BFO between the boundary and the LAO surface.

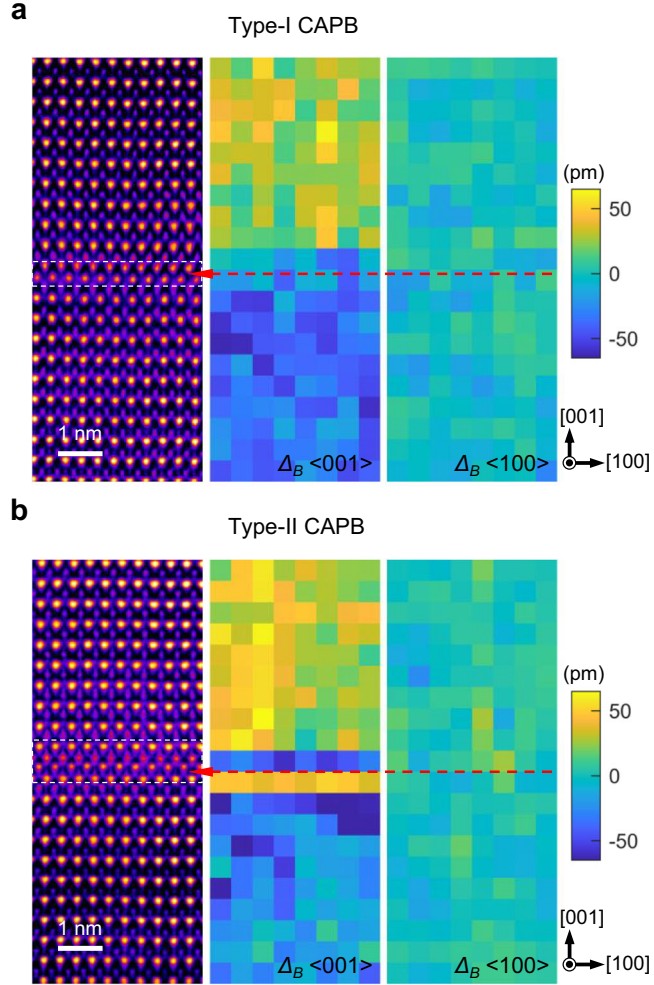

**a** Type-I CAPB

$\Delta_B$ <001>  $\Delta_B$ <100>

(pm)
50
0
-50

[001]
[100]

**b** Type-II CAPB

$\Delta_B$ <001>  $\Delta_B$ <100>

(pm)
50
0
-50

[001]
[100]

**Fig. 4 | Polar features in IP-CAPBs.** Atomic-resolution HAADF images are false-color coded for the easy distinguishment of **a** type-I and **b** type-II CAPBs and the boundaries are marked by white dashed frames. The projected polarization of each u.c. is derived through the displacement components of B-site columns away from the center of corresponding Bi rectangle, $\Delta_B$, along <001> and <100> directions on the right panels. Atomic column shifts pointing to the top in [001] or to the right in [100] directions are defined as positive values, and vice versa. The red arrows indicate the antiphase interfaces in the images and the corresponding polarization maps.

Besides the stacking faults (pointed out by red zigzag lines) along the antiphase boundary, CAPB also features a chemical ordering of anti-site occupation in opposite domains as visualized by the atomic-resolution EDS mapping shown in Fig. 2. Integrating Bi-$M$, Fe-$K$, O-$K$ excitation peaks generates a binary contrast of atomic columns of the respective element, with or without the specific element, in a chemically selective manner. There are three kinds of bonding configurations found in the IP-CAPB: type-I Bi-bilayer, type-II Bi-tetralayer and type-III Fe-bilayer types as listed in Fig. 2a–c, respectively. The excellent anti-site occupation in opposite BFO domains can be well distinguished. The type-II CAPB with Bi-ion tetralayer in the interface can be regarded as a heterostructure of BFO-single u.c. $Bi_2O_3$-BFO, though the normal tetragonal $Bi_2O_3$ phase is a metastable non-polar semiconductor[43]. We also notice that the type-III CAPB involves the direct face-to-face connection of $FeO_6$ octahedra and thus exhibits a high-energy state, making the Fe-bilayer interface rarely observed and spatially discontinuous.

It has been reported that a charged domain wall tends to bend its propagation path to reduce the net bound charge accumulated from the normal polarization of neighboring domains[21]. Our IP-CAPB is also not completely straight as shown in the large-scale image in Fig. 3a. As shown by the geometric phase analysis (GPA) of atomic-resolution image in Fig. 3b, the antiphase configuration is maintained in regardless of the bending. The in-plane π phase variation and half-unit-cell lattice displacement is kept (represented by the compressed in-plane strain $\varepsilon_{xx}$ and the rotation component $\varepsilon_{xy}$) even when the CAPB bends, while the out-of-plane lattice expansion (represented by the tensile out-of-plane strain $\varepsilon_{yy}$) can only be tracked in the bending section, which is explained by the configuration switch between type-I Bi-bilayer and type-II Bi-tetralayer in the bending antiphase interface. From the atomic-resolution EDS mapping of this bending region in Fig. 3c, the alternating interchange of type-I and type-II bonding configurations of a few unit cells makes up of the bending boundary. The type-II CAPB is formed naturally in the overlapped region of two closest neighboring type-I CAPBs. Although the bending behaviors actually imply the bi-stable and switchable states of IP-CAPBs when electric field applied, the propagation direction change does not affect the head-to-head OOP polarization aside the antiphase interface and the charged state of domain walls. Therefore, the question remains why such IP-CAPB bends, even though the accumulated charge and built-up electrostatic energy cannot be released through bending.

The unit-cell-by-unit-cell analysis of polarization in type-I and type-II IP-CAPBs tells us the reason. In Fig. 4, the polarization of each u.c. is divided into orthogonal components along <001> and <100> directions, and quantified by the B-site displacement away from the u.c. center. It is self-evident that either CAPB manifest the atomically sharp head-to-head 180° polarization change across the boundary region (framed by the white dashed rectangle) and the similar polarization strength in BFO domains. As shown in Fig. 5, the local polarization mapping around the same boundary based on the HAADF-STEM imaging clearly demonstrates the head-to-head and charged states of the boundary. The type-II CAPB, however, contains one-unit-cell thick $Bi_2O_3$ inside the wider domain wall. Such $Bi_2O_3$ layer is spontaneously anti-polarized against the surrounding BFO and induces the one-unit-cell thick BFO underneath to switch its polarization direction by 180°, forming an additional tail-to-tail charged boundary that is negatively charged. The creation of antipolar $Bi_2O_3$ layer and the resulted head-to-head/tail-to-tail oppositely charged boundaries may compensate the electrostatic potential built up by single charged domain walls (type-I CAPB), corroborating the stabilization mechanism of IP-CAPBs through bending. Combining these atomic observations and the switching experiments performed on the implanted BFO film as shown in Supplementary Fig. S7, the He-implanted BFO is still ferroelectric and switchable.

While the direct evidence to explain how the He-ion implantation produces IP-CAPBs remains unclear, it has been observed that the oxygen-vacancy density (the average Fe oxidation state) on the top side of IP-CAPBs is higher (lower) than that on the bottom side. This difference has been demonstrated by the lower chemical shift of Fe-$L_{3,2}$ onset and the larger $L_3/L_2$ ratio in the top region by the atomic-resolution electron energy-loss spectroscopy (EELS) as shown in Fig. 6[44]. Because an antiphase domain is actually a region of anti-site defects in the parent lattice, we thus propose that the switch of A- and B-site ions, leading to anti-site defects, was caused by the high-energy ion bombardment and assisted by the generated oxygen vacancies. The accumulation and migration of oxygen vacancies and anti-site defects finally compose the IP-CAPB at the implanted depth after the implantation process[45]. The He-implantation-induced IP-CAPB is reproducible as shown in Supplementary Fig. S8, where we fabricated three batches of samples using the same implantation condition and observed similar IP-CAPBs by cross-sectional TEM. Despite the unclear interacting model between high-energy ions and strongly correlated

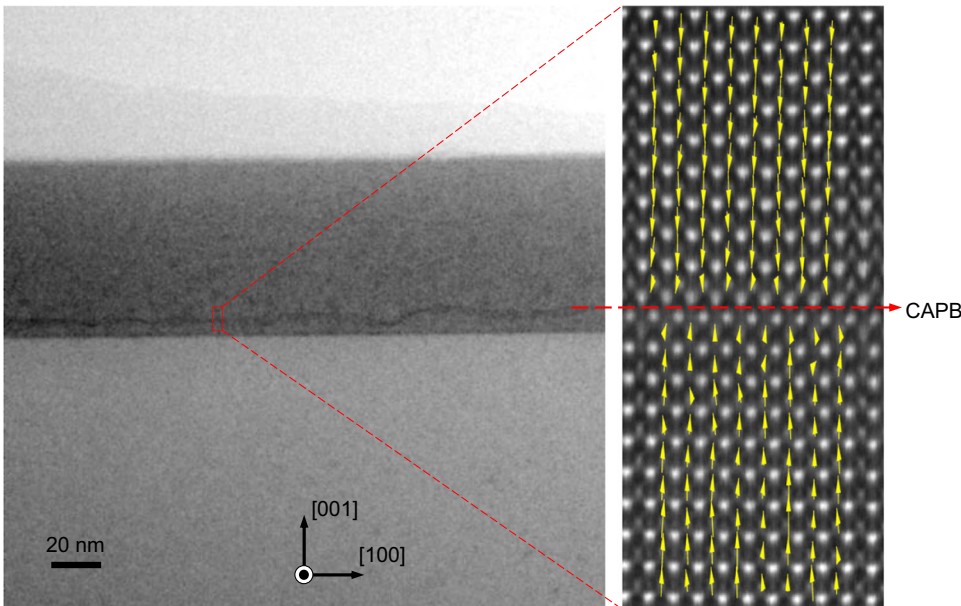

**Fig. 5 | Local polarization mapping around the boundary.** The CAPB region in the red dashed rectangle is zoomed-in to demonstrate the charged state of the boundary. The unit-cell-by-unit-cell polarization is proportionally represented by yellow vectors pointing from the B-site columns to the centers of corresponding A-site rectangles.

crystals, our discovery of the unique IP-CAPB, which possesses an atomically sharp 180° charged interface stabilized by antiphase configuration, provides an unusual playground for studying unconventional domain wall properties and mechanisms, such as the CAPB-controlled magnetoelectric coupling and the electron-hole pair dynamics in a charged domain-wall sequence[4]. Since the IP-CAPB is charged and possibly electrically conductive, its electronic properties can be probed by cross-sectional conductive AFM[22] and exploited in multiferroic tunnel junction devices[23].

To sum up, we demonstrate the facile fabrication of continuous IP-CAPBs in BFO thin films using the He-ion implantation process. This technique is highly scalable, making it suitable for wafer-size lateral patterning. PFM data confirm the disappearance of stripe domain wall structures near the surface of films. Cross-sectional STEM and EDS spectroscopy then reveal a variety of atomic bonding configurations at the implanted depth, where they interweave to create the antiphase interface. Besides the atomically sharp 180° polarization reversal across the IP-CAPB, a single-unit-cell thick antipolar $Bi_2O_3$ resides in the bending region of CAPB, relieving the built-up electrostatic energy. Therefore, this work not only proposes a domain-wall fabrication strategy by the He-ion implantation, but also provides direct atomic-scale observation on such IP-CAPBs for their future utilization in domain-wall nanoelectronics.

## Methods
### Materials synthesis
A series of epitaxial ~70-nm-thick BFO films on LAO (001) substrates were grown by the pulsed laser deposition (PLD) technique at 680 °C under an $O_2$ pressure of 100 mTorr and cooled under 1 atm oxygen atmosphere. After deposition, the He-ion implantation was carried out at 8 keV with a fluence of $2.5 \times 10^{15}$ ions per $cm^2$ at the Ion Beam Center (IBC) at the Helmholtz–Zentrum Dresden–Rossendorf.

### Atomic force microscopy (AFM)
The topographical measurement was conducted by the tapping mode of our Scanning Probe Microscope (model: Asylum Research Cypher) using Al-coated Si probes.

### Piezoresponse force microscopy (PFM)
The PFM measurement was conducted by the vector PFM mode of our Scanning Probe Microscope (model: Asylum Research Cypher) using Pt-coated conductive probes. The cantilever of probes is coated with Al to enhance the reflectivity. During scanning, the tips were applied with 800 mV AC voltage for reading the piezoresponse of BFO films.

### X-ray characterization
The XRD and RSM data, including 2theta-omega line scan and 3D-RSM, were obtained from the six-circle diffractometer equipped with both 0D and 2D detectors in the 1W1A beamline station of the Beijing Synchrotron Radiation Facility using the X-ray wavelength of ~1.548 Å. The 2D RSM images were extracted from the 3D-RSM dataset using MATLAB.

### Electron microscopy and spectroscopy
Focused ion beam (FIB) techniques by FEI Helios G4 UX were applied to prepare the cross-sectional membranes of BFO/ALO films. Before the ion milling processes, the surface was coated with thin carbon paint and deposited 500-nm-thick Pt protecting straps using a low-energy (2 kV) electron beam. Then thicker Pt protecting straps (1.5 μm thick) were deposited using the ion beam. After lifting out a 1-μm-thick plate of the chosen crystal orientation to a copper finger, step-by-step thinning processes using 30, 5, 2, 1, and 0.5 kV ion beams well minimized the beam damage. STEM imaging and spectroscopy were performed using JEOL JEM ARM 200CF equipped with a cold field emission gun and an ASCOR fifth-order probe corrector. HAADF imaging, STEM-EDS and STEM-EELS experiments were carried out under 200 kV accelerating voltage with a 26 mrad convergence semi-angle and 24-120 pA probe currents for optimal information transfer and minimal electron irradiation damage. The collection semi-angles for ADF signal were set to 68–200 mrad. In STEM-EELS measurements, energy dispersion of 0.25 eV per channel and 91 mrad collection semi-angle of Gatan Enfinium ER spectrometer was set up. Zero-loss peak and core-loss edges were simultaneously recorded in dual EELS mode. The simultaneous ADF signal was acquired with a 93 mrad inner collection semi-angle.

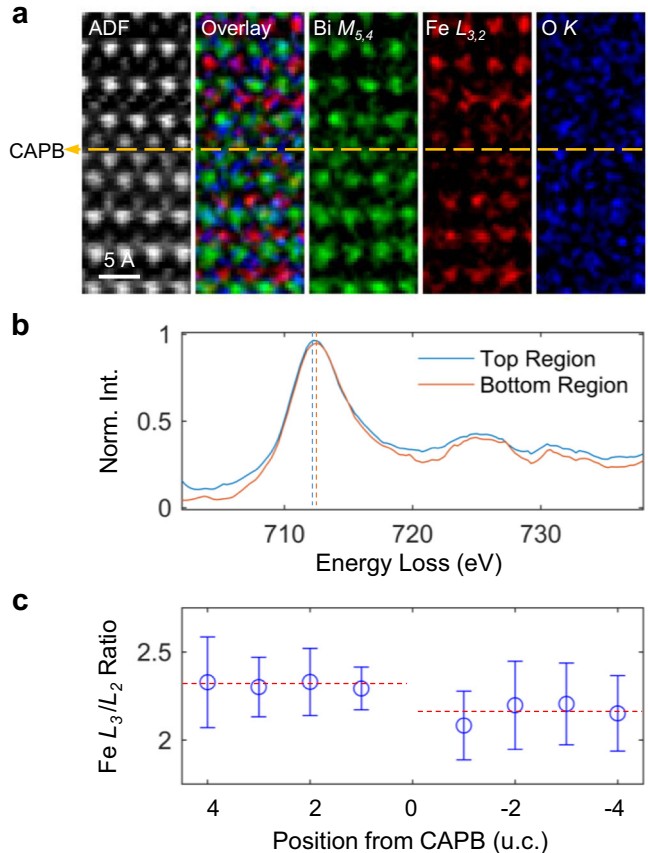

**Fig. 6 | Atomic-resolution electron energy-loss spectroscopy (EELS) analysis across the IP-CAPB in He-implanted BFO films. a** Simultaneously acquired ADF image, overlaid composition map, and the constituent sub-maps by integrating Bi-$M_{5,4}$, Fe-$L_{3,2}$, O-$K$ edges after the power-law background subtraction, respectively, from left to right. The IP-CAPB is marked by an orange dashed line and viewed along the [010] zone axis. **b** Fe-$L_{3,2}$ spectra averaged from the top and bottom regions of (**a**), showing lower chemical shift of Fe-$L_3$ onset in the top region. The peak positions are guided by dashed lines of respective colors. **c** Calculated $L_3/L_2$ ratios of each unit cell from the top (positive u.c. numbers) to the bottom (negative u.c. numbers). The error bars represent the standard deviation of each u.c. row while the red dashed lines indicate mean values of regions above and beneath the IP-CAPB. The close-to-surface side owns larger $c/a$ ratios, lower Fe oxidation states and more implantation-induced oxygen vacancies.

## Reporting summary

Further information on research design is available in the Nature Portfolio Reporting Summary linked to this article.

## Data availability

The authors declare that all data supporting the findings of this study are available within the paper and its Supplementary information files.

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

## Acknowledgements

D.C. thanks the support from the National Natural Science Foundation of China (Grant Nos. 92163210, 91963102), Guangdong Science and Technology Project (Grant No. 2019A050510036), Guangdong Provincial Key Laboratory of Optical Information Materials and Technology (No. 2017B030301007) and Science and Technology Projects in Guangzhou (202201000008). Y.Z. thanks the financial support from the Research Grants Council of Hong Kong through the Early Career Scheme (Project No. 25301617) and the Hong Kong Polytechnic University grant (Project No. 1-ZE6G). Y.G. thanks the funding from the State Key Laboratory of Nuclear Physics and Technology, Peking University (No. NPT2019ZZ01). S.Z. thanks the financial support by the German Research Foundation (Grant No. ZH 225/10-1). X.C. acknowledges the support from the NTU Presidential Postdoctoral Fellowship (Grant No. 03INS001828C230). We hereby also acknowledge the support from 1W1A station of Beijing Synchrotron Radiation Facility.

## Author contributions

X.C. and D.C. conceived the project. C.C. and D.C. synthesized the films and conducted XRD, AFM and PFM measurements. X.C. performed the FIB sample fabrication, electron microscopy characterization and data analyses. L.X. contributed to the structure simulations. Z.G. and Y.C. helped with the ion implantation simulation. G.Z., X.G., W.G., Y.Z. and J.L. supported the project by improving the experiments and the data analysis. X.C. wrote the manuscripts. C.W., Y.G., U.K. and S.Z. conducted the He implantation experiments. All authors participated in the discussion of results and commented on the paper.

## Competing interests

The authors declare no competing interests.
