## [Peer review file · Nature Communications]

REVIEWER COMMENTS

Reviewer #1 (Remarks to the Author):

The authors X. Cai et al. investigate the in-plane charged antiphase boundaries and 180° domain walls in a ferroelectric BiFeO₃ thin film. The investigations of features of atomic width in ferroic materials is of high interest, related to promising applications as nanoelectronic devices or their cumulative macroscopic response, e.g., for next-generation capacitors or actuators. The authors start with an exciting motivation of the topic, followed by a PFM investigation of their films and a very detailed insights obtained by TEM. Unfortunately, the PFM and XRD investigations contain many major mistakes, and the experimental evidence the authors are presenting does not corroborate the conclusion that "PFM confirms the disappearance of in-plane domain structures and the occurrence of uniform out-of plane polarization across the film" (page 8, line 28). Further the absence of a description of the PFM and XRD technique in the methods parts makes it technically impossible for any reader to reproduce the results, nor for the reviewer to judge the experiment. The lack of experimental evidence on how the He-implantation modifies the ferroelectric domain structure makes the subsequent TEM analysis questionable. It remains unclear to the reviewer until the end of the manuscript what the He is actually doing to the ferroelectric polar order in the film and the reviewer does not see experimental evidence of "a new domain-wall fabrication strategy by the controlled He-ion implantation", the authors are claiming in their conclusion. In addition, the language the authors are using in the TEM part is hard to understand for the reviewer, making it presumable difficult that the paper will reach the large audience of a journal such as Nature Communications. I thus cannot recommend the manuscript for publication.

My specific criticisms are (not rated according to importance).

- 1) Page 3, line 11: It depends on the type of semiconductor, which type of defects accumulate at the head-to-head or tail-to-tail walls.
- 2) Page 5, line 6. The authors claim that the pristine film is dominated by the tetragonal-like phase. It is unclear why the authors make this conclusion. Refinement of the pattern or at least peak indexing should be done to corroborate this conclusion. For the reviewer the data presented by the authors looks like a single (002) peak with a broad right shoulder, which can be interpreted as any crystallographic structure.
- 3) Page 5, line 7. The authors perform "topology measurements by AFM". It is unclear what this measurement is. Do the authors mean "topography"?
- 4) Page 5, line 8. The authors obtain crystal structure information from topology AFM (?), which is topography AFM scans in the opinion of the reviewer. Topographic AFM scans, however, provide information on the roughness of the surface and do not contain any information on the crystal structure. The reviewer cannot judge better what the authors are doing, since the experiments are not described in the methods parts.
- 5) Page 5, line 12: The simplified schematics the authors are presenting (displaying out of plane domains) does not match their experimental results. According to the schematic out-of-plane PFM contrast would be expected. However, as displayed in Figure S2, no out of plane contrast can be observed. Instead, the authors observe in-plane PFM contrast in Figure 1A. Also, in the opinion of the reviewer, the origin of the PFM contrast is highly questionable, since the domain structure does not match the domain structure typically observed in PFM thin films of BFO (compare to work by Trassin or Ramesh). Other origins of the PFM contrast and possible artefacts need to be considered and discussed.
- 6) Page 5, line 19: The authors state that their film displays a true tetragonal state after the He-ion implantation. The reviewer sees that the single peak splits into three peaks in Figure S1. The reviewer does not understand how the authors come to the conclusion that this is a true tetragonal state. Detailed refinement of the XRD patterns would be required with a tetragonal crystal symmetry, to corroborate this finding.
- 7) Page 5, line 20: The authors claim that they find a lattice distortion of 1.28 to 1.27. It is unclear how the authors come to this value. The XRD patterns of the authors further look different compared to the XRD patterns previously published in literature

(<https://onlinelibrary.wiley.com/doi/epdf/10.1002/adfm.201101970>) on BFO thin films.

8) Figure 5, line 27: The OOP-PFM result does not change as a function of the ion bombardment. However, the authors claim that based on these data a "single domain state" is produced in BFO films. The authors need to provide convincing experimental evidence that the He-bombardment changes the domain structure at the surface of the film. The experimental evidence provided now is highly contradictory.

Further, a monodomain state is not an evidence for the head-to-head domain structure, the authors are schematically depicting as "He Implanted" in Figure 1A, since subsurface information on the domain structure would be necessarily required to corroborate the conclusion on the 3D domain structure, the authors are making.

9) The PFM is not described in the methods part. Neither is information available on the XRD part. This makes it impossible to reproduce the data.

10) Page 6, line 1: It is unclear to the reviewer, to which "hypothesis as discussed previously", the authors are referring to.

11) It does not get clear how and if the He-implantation switches the domain structure, making the entire TEM study highly speculative, since the ferroelectric state of the sample, particularly in 3D is not clarified prior to the TEM investigations. The authors should answer fundamental questions on the interaction with He and their material and the impact on the domain structure (non of this is done via the PFM presented by the authors). This also involves further experiments to reveal the domain structure before and after He-implantation in all three dimensions. The authors state themselves that "direct evidence to explain how the He-ion implantation produces IP-CAPBs remains unclear" (page 8, line 10), which is highly contradictory to their conclusion and selling point that they find a "new domain-wall fabrication strategy by the controlled He-ion implantation (page 9, line 1), which is the heart of the manuscript.

12) Also, for Figure 1C many questions remain open: how deep is the penetration depth of the He-implantation? Does the He implant into the structure, and can it be observed by atomic imaging? Does it locally change the structure? Is the penetration depth related to the position of the antiphase interface that is displayed in Figure 1C? The role of the He need to be clarified, to experimentally corroborate statements, such as "180° polarization reversal across the IP-CAPB" or "a new domain-wall fabrication strategy" in the conclusion.

Reviewer #2 (Remarks to the Author):

Cai et al. report the formation of in-plane charged phase boundaries in a BiFeO₃ thin film after He-ion implantation. The topic is of strong current interest, the manuscript is well-structured, and the research adds a new perspective to the field of interface engineering in ferroelectrics. Thus, I feel that the work is adequate for publication in Nature Communications after the authors improved the manuscript. A detailed list of points that should be improved is given below.

- Terminology: There are several shortcomings concerning the terminology. Some examples:

o It is not clear what is meant by "crystal domains" vs. "domains" in the first sentence or by "broken local symmetry" in line 4, page 1. I assume in the latter case the authors mean that locally, i.e., at the position of the domain walls, the symmetry is lower than in the domains.

o Line 13, page 1: "head-to-head polarized domains" do not exist. It is the domain wall that is in a head-to-head state, not the domains as they have, by definition, a uniform polarization orientation.

o Line 14, page 1: "out-of-plane polarized ferroelectric" implies a uniform out-of-plane polarization after, e.g., application of an electric field. What is meant here is an "out-of-plane polarization" (not "polarized"), implying that the polar axis is normal to the surface.

o Line 19, page 19: What is a "charge-carried attribute"?

There are more examples throughout the manuscript and I suggest to carefully check the text for proper use of terminology.

- It is written that charged domain walls perpendicular to the film polarization are barely found. This is true for the as-grown state, but examples exist in capacitor geometries; see, e.g., DOI: 10.1038/nnano.2017.51. For completeness, it would be good to mention this.

- Details of the PFM measurements should be provided in the text or figure caption (i.e., applied voltage applied to tip (or back-electrode?), tip material, ...)

- What is the penetration depth of He atoms at 8 kV? It would be good to present respective simulations, discussing the profile and penetration depth.

- The PFM scans show no contrast after He bombardment, which is interpreted as a single-domain state. However, PFM measures the piezoresponse, not the polarization. How did the authors exclude that the He bombardment did not suppress the piezoelectric behavior and, hence, the PFM contrast. Are the samples still ferroelectric/switchable after He implantation? This should be discussed.

- Line 4/5, page 6: Here, the authors write "domain wall" when meaning the phase boundary. This should not be mixed; please check carefully throughout the manuscript.

- Line 7/8, page 6: It is written that electrons accumulate, leading to a "negatively CAPB". Here, different things are mixed. As far as I understand from the manuscript, the APB is in head-to-head configuration and, thus, positively charged. Note that being "charged" refers to the bound charges. If these are screened, this would be achieved by electrons. The screened wall, however, is then neutral and not negatively charged. Also, where do the screening electrons come from? This should be discussed.

- Line 16, page 6: A dark line is observed at 60 nm. How does this relate to the He implantation energy and the expected profile? Is a different position expected if a different energy is used for implantation? This is important with respect to the engineering and should be discussed or, if possible, even be demonstrated.

- Line 12/13, page 7: It is written that the boundary is "winding" and that there are "bending points". In my opinion, boundaries at the resolution are almost never straight and this seems a little overexaggerated. I suggest to tune this down and just state the boundary is not completely straight. Also, in my opinion, the text would be clearer without phrases like "we are amazed". The results by themselves are really interesting and it is not necessary to sell them.

- I am missing a clear demonstration the boundaries are truly charged. Due to the symmetry lowering, all kinds of polarization orientation may arise. The authors should aim for local polarization mapping based on the HAADF-STEM image and confirm that the boundaries are indeed charged as this is essential for the present work.

Reviewer #3 (Remarks to the Author):

The manuscript by Cai et al. reports the experimental investigation of continuous in-plane charged antiphase boundaries (IP-CAPBs) in BiFeO₃ (BFO) thin films prepared by He-ion implantation. Specifically, the in-plane domain structures in as-grown films disappear after the implantation process and are replaced by a uniform out-of-plane domain with different crystalline phase, as shown by piezo-force microscopy and X-ray diffraction. More importantly, careful scanning transmission electron microscopy (STEM) and spectroscopy reveal an antiphase interface with sharp 180° polarization reversal around the implanted depth. While the mechanism of forming such IP-CAPBs is not fully understood, electron energy-loss spectroscopy (EELS) result suggests that the accumulation and migration of oxygen vacancies might be responsible for the structural and electrical reconstruction.

The quality of the experimental data is impressive. The results represent a new route to engineer functional ferroelectric domain walls by He-ion implantation, which is important for future nanoelectronic applications. I can certainly recommend its publication in Nature Communications, provided that the authors take into account my minor comments below.

1. The authors claim that the construction of IP-CAPB is achieved by a "controlled" He-ion implantation technique. However, throughout the entire manuscript, they only report one particular configuration of the BFO thin film and one He-ion dosage. Have they systematically studied the dependence of the interface depth on the implantation voltage / intensity? How did they control the process, if possible at all? In fact, the antiphase boundary seems to be pushed very close to the BFO/LAO interface. It is not clear, from this single experiment, how repeatable or useful this method can be. Please comment on these points.

2. The buried IP-CAPBs with charged interface are likely to be electrically conductive. While further investigation of the electronic properties is beyond the scope of this work, the authors should at least discuss the possible experiment to characterize such sub-surface conductive plane and some scenarios for device applications, which would elevate the significance of the work.

(Note: the responses to reviewers' comments are in blue, the changes made to the manuscript and the supplementary information are highlighted by yellow background in respective documents, also copied here in red for reference.)

Reviewer #1 (Remarks to the Author):

The authors X. Cai et al. investigate the in-plane charged antiphase boundaries and 180° domain walls in a ferroelectric BiFeO_3 thin film. The investigations of features of atomic width in ferroic materials is of high interest, related to promising applications as nanoelectronic devices or their cumulative macroscopic response, e.g., for next-generation capacitors or actuators. The authors start with an exciting motivation of the topic, followed by a PFM investigation of their films and a very detailed insights obtained by TEM.

Response: Thanks a lot for your positive recognition of our atomic insights on this novel in-plane charged antiphase boundary (IP-CAPB).

Unfortunately, the PFM and XRD investigations contain many major mistakes, and the experimental evidence the authors are presenting does not corroborate the conclusion that “PFM confirms the disappearance of in-plane domain structures and the occurrence of uniform out-of-plane polarization across the film” (page 8, line 28).

Response: We have performed more PFM and XRD experiments to corroborate our statements. In the following point-to-point responses, we provide more analyses and appropriate modifications according to the reviewer's comments.

Further the absence of a description of the PFM and XRD technique in the methods parts makes it technically impossible for any reader to reproduce the results, nor for the reviewer to judge the experiment.

Response: Thank you for the comment and we apologize for the careless omission. We have added the description of PFM and XRD techniques in the Methods section in page 10 lines 1-10 of the revised manuscript, which is also copied as below:

“Piezo-response Force Microscopy (PFM): The PFM measurement was conducted by the vector PFM mode of our Scanning Probe Microscope (model: Asylum Research Cypher) using Pt-coated conductive probes. The cantilever of probes is coated with Al to enhance the reflectivity. During scanning, the tips were applied with 800 mV AC voltage for reading the piezo-response of BFO films.

X-ray Characterization: The X-Ray Diffraction (XRD) and Reciprocal Space Map (RSM) data, including 2theta-omega line scan and 3D-RSM, were obtained from the six-circle diffractometer equipped with both 0D and 2D detectors in the 1W1A beamline station of the Beijing Synchrotron Radiation Facility using the X-ray wavelength of $\sim 1.548 \text{ \AA}$. The 2D RSM images were extracted from the 3D-RSM dataset using MATLAB.”

The lack of experimental evidence on how the He-implantation modifies the ferroelectric domain structure makes the subsequent TEM analysis questionable. It remains unclear to the reviewer until the end of the manuscript what the He is actually doing to the ferroelectric polar order in the film and the reviewer does not see experimental evidence of “a new domain-wall fabrication strategy by the controlled He-ion implantation”, the authors are claiming in their conclusion.

Response: In this work, we show the ferroelectric-domain-modification effect of He implantation by comparing the PFM and cross-sectional TEM results of the same BFO/LAO film before and after the He-implantation treatment. As shown in Figure R1 below, the in-plane PFM of pristine and implanted samples clearly demonstrates that the stripe domain wall structures (which is the intrinsic ferroelectric domain structure as reported before, e.g. in *Advanced Functional Materials* 2011, 21, 133-138) near the surface of films disappear after the He implantation, while the out-of-plane PFM components of pristine and implanted samples are similar due to the similar downward polarization in the surface layer of both pristine and implanted samples. Since PFM mainly probes the ferroelectric domain structure near the surface, we further examine the domain structure inside films by cross-sectional TEM techniques as shown in Figure 1 (C) and (D) of the manuscript, by which a continuous in-plane charged antiphase boundary (IP-CAPB) was found to form at the implantation depth, exhibiting 180° domain wall properties. Combining both techniques, we claim that our He-implantation process transforms the stripe domain structure of pristine BFO to a single downward polarized domain near the surface of films and creates a continuous IP-CAPB at the implantation front. Such ferroelectric-domain-modification effect of He implantation is illustrated by the refined schematics as shown in Figure R2 below. Therefore, our results suggest a new domain-wall fabrication strategy by the He implantation.

Because the focus of this work, as reflected by the title “In-plane charged antiphase boundary and 180° domain wall in a ferroelectric film”, is on the first and direct atomic-scale observation of this unique IP-CAPB and the polar order near the boundary by advanced TEM techniques, to provide fundamental atomic insights for their future utilization in domain-wall electronics, we put the mechanism discussion about how the He implantation generates such IP-CAPB at the end of the manuscript.

Therefore, to make the ferroelectric-domain-modification effect of He implantation clearer, we replaced the schematics in Figure 1A with Figure R2 in page 15 of the revised manuscript, and updated the PFM data in Figure 1A and Figure S2 in page 3 of the revised supplementary information with the high-resolution ones of Figure R1. The captions were adjusted accordingly. We also added some explanation to highlight our experimental methods at the beginning of our results in page 4 lines 21-24 of the revised manuscript as “The ferroelectric-domain-modification effect of He implantation is demonstrated by comparing the piezoresponse force microscopy (PFM) and cross-sectional TEM results of the same BFO/LAO film before and after the He-implantation treatment”. Finally, we improved the Abstract in page 2 line 9-14 of the revised manuscript as “High-resolution piezoresponse force microscopy (PFM) results confirm the disappearance of in-plane ferroelectric domain structure and the persistence of a uniform out-of-plane polarization near the surface of film. Then cross-sectional scanning transmission electron microscopy (STEM) and spectroscopy reveal the creation of a continuous IP-CAPB around the implanted depth and a variety of atomic bonding configurations at the antiphase interface”.

Figure R1: High-resolution AFM and PFM characterization of BFO/LAO films (A) before and (B) after the He-ion implantation. The same type of maps in (A) and (B) were plotted in the same color scale for easier comparison.

Figure R2: Schematics illustrating the domain structures in BFO films on LAO substrates before and after the He-ion implantation. The LAO substrates are in gray while the different domain colors in BFO films before and after the He-ion implantation represent various polarization vectors as sketched in corresponding cuboids on the left.

In addition, the language the authors are using in the TEM part is hard to understand for the reviewer, making it presumable difficult that the paper will reach the large audience of a journal such as Nature Communications.

Response: To polish our language, the revised manuscript was proof-read by native speakers and TEM experts, hoping the current version can better present our exciting scientific results.

I thus cannot recommend the manuscript for publication. My specific criticisms are (not rated according to importance).

1) Page 3, line 11: It depends on the type of semiconductor, which type of defects accumulate at the head-to-head or tail-to-tail walls.

Response: We agree with the reviewer that the defect types in semiconductors accumulating around the domain wall can affect its charge state. So we have added this possibility in page 3 line 13-14 of the revised manuscript as “..., while the defect types accumulating around may affect the charge states”.

2) Page 5, line 6. The authors claim that the pristine film is dominated by the tetragonal-like phase. It is unclear why the authors make this conclusion. Refinement of the pattern or at least peak indexing should be done to corroborate this conclusion. For the reviewer the data presented by the authors looks like a single (002) peak with a broad right shoulder, which can be interpreted as any crystallographic structure.

Response: Thank you for the comment. To better verify the phase structure of our pristine BFO, we conducted the X-ray diffraction 2theta-omega scan using the synchrotron beamline. As shown in Figure R3 below, the updated XRD data of pristine BFO show a higher quality, exhibiting the strong (002) M_C (tetragonal-like phase) and weak (002) M_A (rhombohedral-like phase) peaks near the (002) LAO substrate peak, according to the assignment of ref. 38 in the manuscript. Therefore, we claim that the pristine BFO film is dominated by the tetragonal-like phase, actually mixed with small proportion of rhombohedral-like phase, in consistence with the references (ref. 37; ref. 38; Physics Review B 2012, 85, 1, 014104; Physics Review B 2011, 84, 9, 094116).

We thus replaced the Figure S1 in page 2 of the revised supplementary information with Figure R3 of higher quality. We also added more description in page 5 line 9-10 of the revised manuscript as “..., exhibiting the strong (002) M_C (tetragonal-like phase) and weak (002) M_A (rhombohedral-like phase) peaks near the (002) LAO substrate peak³⁷⁻⁴⁰”. Two references on the XRD study of pristine BFO on LAO were included as ref. 39 and ref. 40 here.

Figure R3: 2theta-omega scans of pristine (black) and He-implanted (red) BFO films.

3) Page 5, line 7. The authors perform “topology measurements by AFM”. It is unclear what this measurement is. Do the authors mean “topography”?

Response: Thanks for pointing out this typo and we apologize for any confusion caused. It should be spelt as “topography”, and has been carefully checked and corrected across the manuscript.

4) Page 5, line 8. The authors obtain crystal structure information from topology AFM (?), which is topography AFM scans in the opinion of the reviewer. Topographic AFM scans, however, provide information on the roughness of the surface and do not contain any information on the crystal structure. The reviewer cannot judge better what the authors are doing, since the experiments are not described in the methods parts.

Response: Sorry for the typo again and thank you for giving us the chance to clarify the statement. We agree with the reviewer that the topographic AFM provides information on the roughness or the local height of the surface. As demonstrated in the cited ref. 37 and ref. 38, rhombohedral-like (R) phases come from the distortion of parent matrix to exhibit a smaller c parameter, manifesting lower local height in the topographic AFM map. We can thus ascribe such dark patch contrast to the rhombohedral-like (R-like) phase, which practice was also adopted in the cited ref. 37 and ref. 38. The mixing structure of tetragonal-like and rhombohedral-like phases in the as-grown BFO on LAO is known to be rather complicated stripe domains as you can see from the high-resolution AFM and PFM data in Figure R1 (A), which is not the topic of our work and has been studied in depth in the cited references (e.g. ref. 37; ref. 38; Physics Review B 2012, 85, 1, 014104; Physics Review B 2011, 84, 9, 094116).

To avoid confusion, we added more discussion in page 5 line 14-15 of the revised manuscript as “Because R-like phases come from the distortion of parent matrix to exhibit a smaller c parameter,

manifesting lower local height in the topographic AFM map^{37,38}.”. We also added the description of AFM technique in the Methods section in page 9 lines 26-28 of the revised manuscript, which is also copied as below:

“Atomic Force Microscopy (AFM): The topographical measurement was conducted by the tapping mode of our Scanning Probe Microscope (model: Asylum Research Cypher) using Al-coated Si probes.”

5) Page 5, line 12: The simplified schematics the authors are presenting (displaying out of plane domains) does not match their experimental results. According to the schematic out-of-plane PFM contrast would be expected. However, as displayed in Figure S2, no out of plane contrast can be observed. Instead, the authors observe in-plane PFM contrast in Figure 1A. Also, in the opinion of the reviewer, the origin of the PFM contrast is highly questionable, since the domain structure does not match the domain structure typically observed in PFM thin films of BFO (compare to work by Trassin or Ramesh). Other origins of the PFM contrast and possible artefacts need to be considered and discussed.

Response: Sorry for the confusion caused by our simplified schematics and we believe that as soon as we updated the high-resolution AFM and PFM images of the pristine BFO/LAO film as shown in Figure R1 (A), the reviewer’s doubts should be relieved. There is no out-of-plane domain in the pristine BFO film. Our pristine BFO films possess stripe domain structure mainly originating from the polarization of M_C phase, which is consistent with previous PFM reports (e.g. ref. 38 of the manuscript). The schematics in Figure 1 (A) of the revised manuscript were updated with the ones as shown in Figure R2.

6) Page 5, line 19: The authors state that their film displays a true tetragonal state after the He-ion implantation. The reviewer sees that the single peak splits into three peaks in Figure S1. The reviewer does not understand how the authors come to the conclusion that this is a true tetragonal state. Detailed refinement of the XRD patterns would be required with a tetragonal crystal symmetry, to corroborate this finding.

Response: Thanks for the question. To better verify the phase structure of our He-implanted BFO, we conducted the Reciprocal Space Map (RSM) shown in Figure R4 below, on top of the X-ray diffraction 2theta-omega scan already shown in Figure R3, using the synchrotron beamline. Figure R4 below shows the RSM images of He-implanted BFO around LAO (002) and (103) diffraction spots. We can see that BFO (002) and (103) spots exhibit an elongated shape in contrast to the M_C -like splitting in the pristine BFO (see ref. 38: Advanced Functional Materials 2011, 21, 133-138). Furthermore, LAO (103) and BFO (103) spots stay at the same Q_y position, demonstrating the strict maintenance of tetragonal epitaxy relation. These results are consistent with the previous report on the true tetragonal transition by He-ion implantation (ref. 35 of the manuscript), proving the symmetry-reduced true tetragonal structure in the He-implanted BFO. The multiple BFO (002) peaks in Figure R3 and the elongated shape of BFO (002) and (103) in Figure R4 may be caused by the varied He-ion densities along the implantation depth direction, leading to a wide distribution of tetragonalities in implanted samples.

Therefore, we added the Figure R4 and related discussion on the true tetragonal symmetry change of He-implanted BFO in page 6 of the revised supplementary information as the Figure S5. We

also referred the added Figure S5 in page 5 line 22-24 of the revised manuscript as “By XRD in Figure S1 and Reciprocal Space Map (RSM) in Figure S5 of SI, we notice that the BFO film transforms into the true tetragonal (T) phase after the He-ion implantation”.

Figure R4: RSM results of the He-implanted BFO film obtained around (A) LAO (002) and (B) (103) spots confirm the transformation into a true tetragonal phase.

7) Page 5, line 20: The authors claim that they find a lattice distortion of 1.28 to 1.27. It is unclear how the authors come to this value. The XRD patterns of the authors further look different compared to the XRD patterns previously published in literature (<https://onlinelibrary.wiley.com/doi/epdf/10.1002/adfm.201101970>) on BFO thin films.

Response: Thanks for the comments and these are some misunderstanding.

In the manuscript, we wrote “By XRD in Figure S1 of SI, we notice that the BFO film transforms into the true tetragonal (T) phase after the He-ion implantation with two sub-peaks corresponding to the enhanced c/a ratios of around 1.28 and 1.27.”, not claiming “a lattice distortion of 1.28 to 1.27” as misunderstood by the reviewer. The c/a ratios of 1.28 and 1.27 were obtained by the fitting of atomic images around the boundary as shown in the c/a Ratio plot of Figure 1 (D) in page 15 of the manuscript. We previously tried to assign the splitting (002) XRD peaks of He-implanted BFO to these two observed c/a ratios above and below the boundary, respectively. But we noticed that we claimed more than what we could do here since we cannot get the a -parameter from the XRD in Figure S1 of supplementary information. Therefore, we have modified the statement in page 5 line 22-24 of the revised manuscript as “By XRD in Figure S1 and Reciprocal Space Map (RSM)

in Figure S5 of SI, we notice that the BFO film transforms into the true tetragonal (T) phase after the He-ion implantation”.

Regarding the “XRD patterns look different” comment: firstly, the XRD patterns in the mentioned paper were plotted against the reciprocal L axis while our XRD data were plotted against the real-space 2θ coordinate; secondly, in these two works, the substrates and as-grown BFO phases are different. Different substrates can impose different strain conditions onto the epitaxial growth, leading to distinct phases of different lattice parameters in BFO. In the mentioned paper, the R phase is the dominant phase grown on STO substrates while the T phase is induced by secondary phase-transition processes. But in our work, the R-like phases are incompletely relaxed from the dominant T-like phase on LAO substrates. Therefore, it’s evident that the XRD patterns should be different.

8) Figure 5, line 27: The OOP-PFM result does not change as a function of the ion bombardment. However, the authors claim that based on these data a “single domain state” is produced in BFO films. The authors need to provide convincing experimental evidence that the He-bombardment changes the domain structure at the surface of the film. The experimental evidence provided now is highly contradictory.

Further, a monodomain state is not an evidence for the head-to-head domain structure, the authors are schematically depicting as “He Implanted” in Figure 1A, since subsurface information on the domain structure would be necessarily required to corroborate the conclusion on the 3D domain structure, the authors are making.

Response: Sorry for the confusion caused by our simplified schematics and we believe that as soon as we updated the high-resolution PFM images and schematics of the pristine BFO/LAO film as shown in Figure R1 (A) and Figure R2, the reviewer’s doubts should be relieved. In this work, we show the ferroelectric-domain-modification effect of He implantation by comparing the PFM and cross-sectional TEM results of the same BFO/LAO film before and after the He-implantation treatment. Through the combination of surface-probed technique by PFM and cross-section-probed technique by TEM, the ferroelectric-domain modified by He implantation can be revealed in three dimensions.

As shown in Figure R1 (A), the out-of-plane PFM components of pristine and implanted samples are similar due to the similar downward polarization in the surface layer of both pristine and implanted samples. Since PFM mainly probes the ferroelectric domain structure near the surface, we further examine the domain structure inside films by cross-sectional TEM techniques as shown in Figure 1 (C) and (D) in the manuscript, by which a continuous in-plane charged antiphase boundary (IP-CAPB) was found to form at the implantation depth, exhibiting 180° domain wall properties. Combining both techniques, we claim that our He-implantation process transforms the stripe domain structure of pristine BFO to a single downward polarized domain near the surface of films and creates a continuous IP-CAPB at the implantation front. Such ferroelectric-domain-modification effect of He implantation is illustrated by the refined schematics as shown in Figure R2 above.

Therefore, we replaced the schematics in Figure 1A with better-drawn Figure R2 in page 15 of the revised manuscript, and updated the PFM data in Figure 1A in page 15 of the revised manuscript and Figure S2 in page 3 of the revised supplementary information with the high-resolution ones in

Figure R1. To avoid misunderstanding and be rigorous we also modified the statement in page 5 lines 30-31 of the revised manuscript as “...the homogeneous contrast in both IP- and OOP-PFM results indicate the single domain state near the surface of BFO films after implantation”.

9) The PFM is not described in the methods part. Neither is information available on the XRD part. This makes it impossible to reproduce the data.

Response: Thank you for the comment and we apologize for the careless omission. We have added the description of PFM and XRD techniques in the Methods section in page 10 lines 1-10 of the revised manuscript, which is also copied as below:

“Piezo-response Force Microscopy (PFM): The PFM measurement was conducted by the vector PFM mode of our Scanning Probe Microscope (model: Asylum Research Cypher) using Pt-coated conductive probes. The cantilever of probes is coated with Al to enhance the reflectivity. During scanning, the tips were applied with 800 mV AC voltage for reading the piezo-response of BFO films.

X-ray Characterization: The X-Ray Diffraction (XRD) and Reciprocal Space Map (RSM) data, including 2theta-omega line scan and 3D-RSM, were obtained from the six-circle diffractometer equipped with both 0D and 2D detectors in the 1W1A beamline station of the Beijing Synchrotron Radiation Facility using the X-ray wavelength of $\sim 1.548 \text{ \AA}$. The 2D RSM images were extracted from the 3D-RSM dataset using MATLAB.”

10) Page 6, line 1: It is unclear to the reviewer, to which “hypothesis as discussed previously”, the authors are referring to.

Response: The full sentence reads “According to the hypothesis as discussed previously to hybridize the antiphase and the in-plane charged characteristics into a single domain wall, ...”, we are mentioning the desire to hybridize the antiphase and the in-plane charged properties into a single domain wall, as discussed in the previous introduction paragraph in page 4 lines 3-8. To make the text clearer, we modified the sentence as “According to the motivation as discussed previously in the introduction paragraph to hybridize the antiphase and the in-plane charged characteristics into a single domain wall” in page 6 line 1.

11) It does not get clear how and if the He-implantation switches the domain structure, making the entire TEM study highly speculative, since the ferroelectric state of the sample, particularly in 3D is not clarified prior to the TEM investigations. The authors should answer fundamental questions on the interaction with He and their material and the impact on the domain structure (non of this is done via the PFM presented by the authors). This also involves further experiments to reveal the domain structure before and after He-implantation in all three dimensions. The authors state themselves that “direct evidence to explain how the He-ion implantation produces IP-CAPBs remains unclear” (page 8, line 10), which is highly contradictory to their conclusion and selling point that they find a “new domain-wall fabrication strategy by the controlled He-ion implantation (page 9, line 1), which is the heart of the manuscript.

Response: Thanks for the chance for us to clarify the experiments and selling points. In this work, we show the ferroelectric-domain-modification effect of He implantation by comparing the PFM

and cross-sectional TEM results of the same BFO/LAO film before and after the He-implantation treatment. As shown in Figure R1, the in-plane PFM of pristine and implanted samples clearly demonstrates that the stripe domain wall structures (which is the intrinsic ferroelectric domain structure as reported before, e.g. in *Advanced Functional Materials* 2011, 21, 133-138) near the surface of films disappear after the He implantation, while the out-of-plane PFM components of pristine and implanted samples are similar due to the similar downward polarization in the surface layer of both pristine and implanted samples. Since PFM mainly probes the ferroelectric domain structure near the surface, we further examine the domain structure inside films by cross-sectional TEM techniques as shown in Figure 1 (C) and (D) in the manuscript, by which a continuous in-plane charged antiphase boundary (IP-CAPB) was found to form at the implantation depth, exhibiting 180° domain wall properties as demonstrated in Figure 2, 3 and 4 of the manuscript. Through the combination of surface-probed technique by PFM and cross-section-probed technique by TEM, the ferroelectric-domain structure modified by He implantation can be revealed in three dimensions. Therefore, we claim that our He-implantation process transforms the stripe domain structure of pristine BFO to a single downward polarized domain near the surface of films and creates a continuous IP-CAPB at the implantation front. Therefore, our results suggests a new domain-wall fabrication strategy by the He implantation.

The “direct evidence to explain how the He-ion implantation produces IP-CAPBs” usually means the direct observation of the interactions between high-energy He-ion and BFO, requiring an *in-situ* manner, which is impossible for the currently available technology framework. However, through the correlated microscopy by surface-probed PFM and cross-section-probed TEM on the same BFO/LAO film before and after the He-implantation, we can conclude that the He-ion implantation technique provides a new domain-wall fabrication strategy.

12) Also, for Figure 1C many questions remain open: how deep is the penetration depth of the He-implantation? Does the He implant into the structure, and can it be observed by atomic imaging? Does it locally change the structure? Is the penetration depth related to the position of the antiphase interface that is displayed in Figure 1C? The role of the He need to be clarified, to experimentally corroborate statements, such as “180° polarization reversal across the IP-CAPB” or “a new domain-wall fabrication strategy” in the conclusion.

Response: As shown in Figure 1(C) and Figure 3(A), the penetration depth of the He implantation is around 60 nm, corresponding to the position of IP-CAPB. The comparison of XRD, PFM and cross-sectional TEM results of the same BFO/LAO film before and after the He-implantation treatment demonstrates the He implantation and its modification to BFO structure. As shown in Figure 1(D), Figure 2, Figure 3, Figure 4 and Figure 5, He ions cannot be observed by atomic imaging due to its small atomic number (2) in contrast to the heavy Bi atoms (83). As discussed in page 8 line 21-31, it was observed that the oxygen-vacancy density in the top side of IP-CAPBs is higher than that in the bottom side. This difference was also demonstrated by the lower chemical shift of Fe- $L_{3,2}$ onset and the larger L_3/L_2 ratio in the top side by electron energy-loss spectroscopy in Figure 5 of the manuscript. Because an antiphase domain is actually a region of anti-site defects in the parent lattice, we thus propose that the switch of A- and B-site ions, leading to anti-site defects, may be caused by the high-energy ion bombardment and assisted by the generated oxygen vacancies. The accumulation and migration of oxygen vacancies and anti-site defects finally compose the IP-CAPB at the implanted depth after the implantation process. In this work, through the correlated microscopy by surface-probed PFM and cross-section-probed TEM on the same

BFO/LAO film before and after the He-implantation, we can conclude that the He-ion implantation technique provides a new domain-wall fabrication strategy.

Reviewer #2 (Remarks to the Author):

Cai et al. report the formation of in-plane charged phase boundaries in a BiFeO₃ thin film after He-ion implantation. The topic is of strong current interest, the manuscript is well-structured, and the research adds a new perspective to the field of interface engineering in ferroelectrics. Thus, I feel that the work is adequate for publication in Nature Communications after the authors improved the manuscript. A detailed list of points that should be improved is given below.

Response: Thank you so much for the careful review and the positive recommendation on our work. Please find our point-to-point responses as follow. We are so grateful for your important suggestions to improve the paper much!

- Terminology: There are several shortcomings concerning the terminology. Some examples:

o It is not clear what is meant by “crystal domains” vs. “domains” in the first sentence or by “broken local symmetry” in line 4, page 1. I assume in the latter case the authors mean that locally, i.e., at the position of the domain walls, the symmetry is lower than in the domains.

Response: We modified the “crystal domains” as “domains” to avoid confusion and improved the sentence as “In comparison to the three-dimensional domains in ferroelectric thin films, the two-dimensional domain walls or boundaries ... typically have the reduced dimensionality of only a few atomic layers” in page 3 line 1-3 of the revised manuscript. Here we are comparing the crystalline symmetry of domains (three-dimensional) and domain boundaries (two-dimensional) in thin films. Yes, that “These domain walls possess broken local symmetry” means the lowered crystallographic symmetry at the position of domain walls.

o Line 13, page 1: “head-to-head polarized domains” do not exist. It is the domain wall that is in a head-to-head state, not the domains as they have, by definition, a uniform polarization orientation.

Response: Sorry for this confusion caused. We have corrected the phrase as “...a domain wall in the head-to-head (or tail-to-tail) state...” in page 3 line 12-13.

o Line 14, page 1: “out-of-plane polarized ferroelectric” implies a uniform out-of-plane polarization after, e.g., application of an electric field. What is meant here is an “out-of-plane polarization” (not “polarized”), implying that the polar axis is normal to the surface.

Response: Yes, we agree with the reviewer and have corrected the words as “In ferroelectric thin films with out-of-plane polarization...” in page 3 line 14-15.

o Line 19, page 19: What is a “charge-carried attribute”?

Response: Sorry for this inappropriate word. Here we tried to express the properties of being charged in charged domain walls. We modified it as “charged properties” in page 3 line 20.

There are more examples throughout the manuscript and I suggest to carefully check the text for proper use of terminology.

Response: Thanks for pointing out the inappropriate terminology, we have checked and corrected them throughout the manuscript and supplementary information to our best.

- It is written that charged domain walls perpendicular to the film polarization are barely found. This is true for the as-grown state, but examples exist in capacitor geometries; see, e.g., DOI: 10.1038/nnano.2017.51. For completeness, it would be good to mention this.

Response: Yes, we agree with the Reviewer that this example is important to be included here for completeness. We improved the discussion in page 3 line 16-18 as “...a charged domain wall extending perpendicular to the film polarization direction is barely found in as-grown films (although examples exist in capacitor geometries⁴⁵)...” and added the mentioned reference as ref. 45 of the revised manuscript.

- Details of the PFM measurements should be provided in the text or figure caption (i.e., applied voltage applied to tip (or back-electrode?), tip material, ...)

Response: Thank you for the comment and we apologize for the careless omission. We have added the description of PFM measurements in the Methods section in page 10 lines 1-5 of the revised manuscript, which is also copied as below:

“Piezo-response Force Microscopy (PFM): The PFM measurement was conducted by the vector PFM mode of our Scanning Probe Microscope (model: Asylum Research Cypher) using Pt-coated conductive probes. The cantilever of probes is coated with Al to enhance the reflectivity. During scanning, the tips were applied with 800 mV AC voltage for reading the piezo-response of BFO films.

- What is the penetration depth of He atoms at 8 kV? It would be good to present respective simulations, discussing the profile and penetration depth.

Response: Thanks for the good suggestion. As shown in Figure R5 below, we simulated the He-ion implantation depth profiles in BFO under different beam energies by the open-source software SRIM-2013 (URL: <http://www.srim.org/SRIM/SRIMLEGL.htm>). Although the penetration depth of 8 kV He can be as deep as 100 nm, the highest He density only reaches a depth of around 60 nm (the depth where we observed the IP-CAPB under 8 kV beam energy), and then the He density drops very fast, which is consistent with our proposed mechanism how He implantation generates the IP-CAPB in page 8 line 21-31 of the manuscript. It was observed that the oxygen-vacancy density in the top side of IP-CAPBs is higher than that in the bottom side. An antiphase domain is actually a region of anti-site defects in the parent lattice. The switch of A- and B-site ions, leading to anti-site defects, can be triggered by the high-energy He bombardment and favored by the implantation-generated oxygen vacancies. Therefore, the accumulation and migration of oxygen vacancies and anti-site defects to a critical level finally compose the IP-CAPB at the implantation front, where the high-enough He-ion density can arrive. Varied beam energies can lead to different implantation depth and profiles as shown in Figure R5. Since the simulation does not involve the effects caused by the LAO substrate, there may be some slight deviation in the practical depth profiles.

To support our discovery, we added Figure R5 as Figure S6, and the discussion on implantation depth profiles in page 7 of the revised supplementary information.

Figure R5: Simulation of He-ion implantation depth profiles in BFO under different beam energies by the open-source software SRIM-2013 (URL: <http://www.srim.org/SRIM/SRIMLEGL.htm>).

- The PFM scans show no contrast after He bombardment, which is interpreted as a single-domain state. However, PFM measures the piezoresponse, not the polarization. How did the authors exclude that the He bombardment did not suppress the piezoelectric behavior and, hence, the PFM contrast. Are the samples still ferroelectric/switchable after He implantation? This should be discussed.

Response: Thank you for this nice question. As shown in Figure R6 below, we performed the switching experiment on a BFO film implanted in the same way as reported in the main text. The He-ion implanted samples are still switchable. Combining the observed polarization unit-cell-by-unit-cell in implanted BFO films as shown in Figure 4 of the manuscript, we can conclude that the He-implanted BFO is still ferroelectric and switchable.

Therefore, we added Figure R6 as Figure S7 in page 8 of the revised supplementary information, and included more discussion in page 8 line 18-20 of the revised manuscript as “Combining these atomic observation and the switching experiment performed on the implanted BFO film as shown in the Figure S7 of SI, the He-implanted BFO is still ferroelectric and switchable”.

Figure R6: Switching behavior after He-ion implantation. (A) AFM image, (B) out-of-plane PFM phase and (C) amplitude images of a written rectangle in the implanted BFO film. The film was implanted in the same way as reported in the main text with a $\text{Ca}_{0.96}\text{Ce}_{0.04}\text{MnO}_3$ layer grown under BFO film as the back electrode. The field poling was performed by applying -10V DC bias to the conductive PFM probe.

- Line 4/5, page 6: Here, the authors write “domain wall” when meaning the phase boundary. This should not be mixed; please check carefully throughout the manuscript.

Response: Sorry for the mistake. We have corrected the “domain wall” into “boundary” here, and we have checked carefully throughout the manuscript.

- Line 7/8, page 6: It is written that electrons accumulate, leading to a “negatively CAPB”. Here, different things are mixed. As far as I understand from the manuscript, the APB is in head-to-head configuration and, thus, positively charged. Note that being “charged” refers to the bound charges. If these are screened, this would be achieved by electrons. The screened wall, however, is then neutral and not negatively charged. Also, where do the screening electrons come from? This should be discussed.

Response: Sorry for the confusion caused and thank you for pointing out the misuse of concepts. Yes, the APB in our work is in the head-to-head configuration and thus positively charged. We did not observe possible sources of screening electrons. We thus improved the discussion in page 6 line 7-8 of the revised manuscript as “Owing to the head-to-head OOP polarization aside the antiphase interface, the CAPB is positively charged by bound charges, since we did not observe possible sources of screening electrons”.

- Line 16, page 6: A dark line is observed at 60 nm. How does this relate to the He implantation energy and the expected profile? Is a different position expected if a different energy is used for implantation? This is important with respect to the engineering and should be discussed or, if possible, even be demonstrated.

Response: Thanks for the good question. As shown in Figure R5 above, we simulated the He-ion implantation depth profiles in BFO under different beam energies. Although the penetration depth

of 8 kV He can be as deep as 100 nm, the highest He density only reaches a depth of around 60 nm (the depth where we observed the IP-CAPB under 8 kV beam energy), and then the He density drops very fast, which is consistent with our proposed mechanism how He implantation generates the IP-CAPB. We found that the oxygen-vacancy density in the top side of IP-CAPBs is higher than that in the bottom side as shown in Figure 5 of the manuscript. An antiphase domain is actually a region of anti-site defects in the parent lattice. The switch of A- and B-site ions, leading to anti-site defects, can be triggered by the high-energy He bombardment and favored by the implantation-generated oxygen vacancies. The accumulation and migration of oxygen vacancies and anti-site defects to a critical level finally compose the IP-CAPB at the implantation front, where the high-enough He-ion density can arrive. Varied beam energies can lead to different implantation depths and profiles as shown in Figure R5. Thus a different CAPB position can be expected if a different energy is used for the implantation. However, the experimental demonstration of CAPB at varied depths is still under progress and requires much more time for matching schedules of different labs (the He-ion implantation was carried out at the Ion Beam Center (IBC) at the Helmholtz-Zentrum Dresden-Rossendorf). The current manuscript focuses on the first direct atomic-scale observation of this unique IP-CAPB and the polar order near the boundary by advanced TEM techniques, to provide fundamental atomic insights for their future utilization in domain-wall electronics. We hope to present the vertical engineering and lateral patterning of such IP-CAPB for practical device applications in our future work soon.

Therefore, we added more discussion on the IP-CAPB depth in page 6 line 17-23 of the revised manuscript as “This depth position is in consistence with our simulation of He implantation depth profiles as shown in Figure S6 of SI. The accumulation and migration of oxygen vacancies and generated anti-site defects to a critical level by the He implantation compose the IP-CAPB at the implantation front, where the high-enough He-ion density can arrive under 8 kV beam energy. Varied beam energies can lead to different implantation depths and profiles as shown in Figure S6 of SI. Thus a different IP-CAPB position can be expected if a different energy is used for the implantation”.

- Line 12/13, page 7: It is written that the boundary is “winding” and that there are “bending points”. In my opinion, boundaries at the resolution are almost never straight and this seems a little overexaggerated. I suggest to tune this down and just state the boundary is not completely straight. Also, in my opinion, the text would be clearer without phrases like “we are amazed”. The results by themselves are really interested and it is not necessary to sell them.

Response: We appreciate the useful suggestions. We agree with the Reviewer that boundaries are almost never straight and modified the sentence in page 7 line 18-21 of the revised manuscript into “Our IP-CAPB is not completely straight as shown in the large-scale image in Figure 3 (A). As shown by the geometric phase analysis (GPA) of atomic-resolution image in Figure 3 (B), the antiphase configuration is maintained in regardless of the bending”. We also removed the “we are amazed” phrases to make text clearer.

- I am missing a clear demonstration the boundaries are truly charged. Due to the symmetry lowering, all kinds of polarization orientation may arise. The authors should aim for local polarization mapping based on the HAADF-STEM image and confirm that the boundaries are indeed charged as this is essential for the present work.

Response: Thanks for the critical comment. We performed the local polarization mapping around the boundary based on the HAADF-STEM imaging as shown in Figure R7 below, following the method adopted by the ref. 23 (Nature 2023, 613, 656–661). The head-to-head state around the boundary can be clearly seen. So the boundary should be positively charged by bound charges.

Figure R7: Local polarization mapping around the boundary based on the HAADF-STEM imaging.

Therefore, we added Figure R7 as Figure S8 in page 9 of the revised supplementary information, and included more discussion in page 8 line 9-11 of the revised manuscript as “As shown in Figure S8 of SI, the local polarization mapping around the same boundary based on the HAADF-STEM imaging clearly demonstrates the head-to-head and charged states of the boundary”.

Reviewer #3 (Remarks to the Author):

The manuscript by Cai et al. reports the experimental investigation of continuous in-plane charged antiphase boundaries (IP-CAPBs) in BiFeO₃ (BFO) thin films prepared by He-ion implantation. Specifically, the in-plane domain structures in as-grown films disappear after the implantation process and are replaced by a uniform out-of-plane domain with different crystalline phase, as shown by piezo-force microscopy and X-ray diffraction. More importantly, careful scanning transmission electron microscopy (STEM) and spectroscopy reveal an antiphase interface with sharp 180° polarization reversal around the implanted depth. While the mechanism of forming such IP-CAPBs is not fully understood, electron energy-loss spectroscopy (EELS) result suggests that the accumulation and migration of oxygen vacancies might be responsible for the structural and electrical reconstruction. The quality of the experimental data is impressive. The results represent a new route to engineer functional ferroelectric domain walls by He-ion implantation,

which is important for future nanoelectronic applications. I can certainly recommend its publication in Nature Communications, provided that the authors take into account my minor comments below.

Response: Thank you very much for the careful review and such positive recommendation on our work. Please find our point-to-point responses as follow. We again express our sincere gratitude to your valuable suggestions to improve the paper.

1. The authors claim that the construction of IP-CAPB is achieved by a "controlled" He-ion implantation technique. However, throughout the entire manuscript, they only report one particular configuration of the BFO thin film and one He-ion dosage. Have they systematically studied the dependence of the interface depth on the implantation voltage / intensity? How did they control the process, if possible at all? In fact, the antiphase boundary seems to be pushed very close to the BFO/LAO interface. It is not clear, from this single experiment, how repeatable or useful this method can be. Please comment on these points.

Response: Thanks for the critical question. As shown in Figure R5 below, we simulated the He-ion implantation depth profiles in BFO under different beam energies. Under 8 kV He implantation, the highest He density only reaches a depth of around 60 nm (the depth where we observed the IP-CAPB under 8 kV beam energy), and then the He density drops very fast, which is consistent with our proposed mechanism how He implantation generates the IP-CAPB in page 8 line 21-31 of the manuscript. We found that the oxygen-vacancy density in the top side of IP-CAPBs is higher than that in the bottom side as demonstrated in Figure 5 of the manuscript. An antiphase domain is actually a region of anti-site defects in the parent lattice. The switch of A- and B-site ions, leading to anti-site defects, can be triggered by the high-energy He bombardment and favored by the implantation-generated oxygen vacancies. The accumulation and migration of oxygen vacancies and anti-site defects to a critical level finally compose the IP-CAPB at the implantation front, where the high-enough He-ion density can arrive. Therefore, varied beam energies can lead to different implantation depths and profiles as shown in Figure R5. A different CAPB position can be expected if a different energy is used for the implantation. However, the experimental demonstration of CAPB at varied depths is still under progress and requires much more time for matching schedules of different labs (the He-ion implantation was carried out at the Ion Beam Center (IBC) at the Helmholtz-Zentrum Dresden-Rossendorf). The current manuscript focuses on the first direct atomic-scale observation of this unique IP-CAPB and the polar order near the boundary by advanced TEM techniques, to provide fundamental atomic insights for their future utilization in domain-wall electronics. We hope to present the vertical engineering and lateral patterning of such IP-CAPB for practical device applications in our future work soon.

Regarding the repeatability of this method, we fabricated three batches of samples using the same implantation condition and observed similar IP-CAPBs by cross-sectional TEM as shown in Figure R8 below, demonstrating the reasonable reproducibility of IP-CAPBs under this He-implantation process.

To support our discovery, we added Figure R5 as Figure S6, and the discussion on manipulating boundary depth in page 7 of the revised supplementary information. We also removed the possibly over-claimed "controlled" statements throughout the manuscript. Finally, we added Figure R8 as Figure S9 in page 10 of the revised supplementary information and added the repeatability discussion in page 8 line 30-31 of the revised manuscript as "The He-implantation-induced IP-

CAPB is reproducible as shown in Figure S9 of SI, where we fabricated three batches of samples using the same implantation condition and observed similar IP-CAPBs by cross-sectional TEM”.

Figure R5: Simulation of He-ion implantation depth profiles in BFO under different beam energies by the open-source software SRIM-2013 (URL: <http://www.srim.org/SRIM/SRIMLEGL.htm>).

Figure R8: Reproducibility of IP-CAPBs.

2. The buried IP-CAPBs with charged interface are likely to be electrically conductive. While further investigation of the electronic properties is beyond the scope of this work, the authors should at least discuss the possible experiment to characterize such sub-surface conductive plane and some scenarios for device applications, which would elevate the significance of the work.

Response: Thank you for the useful advice to improve the significance of this work. We fully agree with the Reviewer that this IP-CAPB can embark exciting investigation on its rich electronic properties. Some discussion was included in page 9 line 6-8 of the revised manuscript, which was copied here as “Since the IP-CAPB is charged and possibly electrically conductive, its electronic

properties can be probed by cross-sectional conductive AFM⁴⁵ and exploited in multiferroic tunnel junction devices⁴⁶”. Two references about the characterization of sub-surface conductive plane and the scenario for device applications were added as ref. 45 and ref. 46 of the revised manuscript, to inspire interested readers.

Other changes made to the revised manuscript:

1. The author list in page 1 line 3-4 of the revised manuscript was modified by adding Zixin Gui and Yu Chen as coauthors, in recognition of their contribution to the extra data provided in this revision. The author-affiliation labelling was accordingly adjusted. All other authors agreed on the addition and the order adjustment of authorship.
2. The original reference labelling was accordingly adjusted in compliance with the insertion of new references.
3. The acknowledgement towards synchrotron beamline services was added in page 11 line 9-10.
4. A “Data Availability” section was added in page 11 line 20-22 of the revised manuscript, and the Table of Contents Graphic was removed to follow the journal style.

REVIEWERS' COMMENTS

Reviewer #2 (Remarks to the Author):

The authors have responded to all my comments and provided additional data to corroborate their claims. The only minor change I suggest is to show Figure R7 already in the main text. I feel that this is an important Figure, showing nice and highly relevant data.

I recommend to publish the revised manuscript in Nature Communications.

Reviewer #3 (Remarks to the Author):

In the revised manuscript, the authors have added new information to the SI and new references to the main text. I am happy to see that my previous comments and concerns are satisfactorily addressed. I can recommend its publication as is.

(Note: the responses to reviewers' comments are in blue, the changes made to the manuscript and the supplementary information are highlighted by yellow background in respective documents, also copied here in red for reference.)

Reviewer #2 (Remarks to the Author):

The authors have responded to all my comments and provided additional data to corroborate their claims. The only minor change I suggest is to show Figure R7 already in the main text. I feel that this is an important Figure, showing nice and highly relevant data.

I recommend to publish the revised manuscript in Nature Communications.

Response: Thank you so much for your critical advice and recommendation. We moved the Figure R7 into the revised main text as Figure 5 in page 19, and modified the relevant discussion in page 8 line 11-13 of the revised manuscript as “As shown in Figure 5, the local polarization mapping around the same boundary based on the HAADF-STEM imaging clearly demonstrates the head-to-head and charged states of the boundary”. The labeling of other figures was then adjusted to accommodate this change.

Figure 5: Local polarization mapping around the boundary.

Reviewer #3 (Remarks to the Author):

In the revised manuscript, the authors have added new information to the SI and new references to the main text. I am happy to see that my previous comments and concerns are satisfactorily addressed. I can recommend its publication as is.

Response: Thank you so much for your valuable recommendation.